

# The effect of modulating the quantity of enzymes in a model ethanol pathway on metabolic flux in *Synechocystis* sp. PCC 6803

Paulina Bartasun[1], Nicole Prandi[1], Marko Storch[1,2], Yarin Aknin[1,3], Mark Bennett[1], Arianna Palma[1], Geoff Baldwin[1,2], Yumiko Sakuragi[4], Patrik R. Jones[1,2] and John Rowland[1]

[1] Department of Life Sciences, Imperial College London, London, United Kingdom
[2] Imperial College Centre for Synthetic Biology, Imperial College London, London, United Kingdom
[3] Institute of Plant Sciences and Genetics in Agriculture, Hebrew University of Jerusalem, Rehovot, Israel
[4] Department of Plant and Environmental Sciences, University of Copenhagen, Frederiksberg, Denmark

Corresponding author
Patrik R. Jones,
p.jones@imperial.ac.uk

## ABSTRACT

Synthetic metabolism allows new metabolic capabilities to be introduced into strains for biotechnology applications. Such engineered metabolic pathways are unlikely to function optimally as initially designed and native metabolism may not efficiently support the introduced pathway without further intervention. To develop our understanding of optimal metabolic engineering strategies, a two-enzyme ethanol pathway consisting of pyruvate decarboxylase and acetaldehyde reductase was introduced into *Synechocystis* sp. PCC 6803. We characterised a new set of ribosome binding site sequences in *Synechocystis* sp. PCC 6803 providing a range of translation strengths for different genes under test. The effect of ribosome-binding site sequence, operon design and modifications to native metabolism on pathway flux was analysed by HPLC. The accumulation of all introduced proteins was also quantified using selected reaction monitoring mass spectrometry. Pathway productivity was more strongly dependent on the accumulation of pyruvate decarboxylase than acetaldehyde reductase. In fact, abolishment of reductase over-expression resulted in the greatest ethanol productivity, most likely because strains harbouring single-gene constructs accumulated more pyruvate decarboxylase than strains carrying any of the multi-gene constructs. Overall, several lessons were learned. Firstly, the expression level of the first gene in any operon influenced the expression level of subsequent genes, demonstrating that translational coupling can also occur in cyanobacteria. Longer operons resulted in lower protein abundance for proximally-encoded cistrons. And, implementation of metabolic engineering strategies that have previously been shown to enhance the growth or yield of pyruvate dependent products, through co-expression with pyruvate kinase and/or fructose-1,6-bisphosphatase/sedoheptulose-1,7-bisphosphatase, indicated that other factors had greater control over growth and metabolic flux under the tested conditions.

## INTRODUCTION

Synthetic metabolism is the creation of novel metabolism achieved by the combination of enzymes and regulatory elements that in most cases have not co-evolved. Although this opens up new opportunities for biotechnological solutions (*Erb, Jones & Bar-Even, 2017*) such synthetic pathways are unlikely to function optimally due to sub-optimal balance of enzyme abundance and/or regulation of expression. For example, *Zelcbuch et al. (2013)* clearly showed the impact of variation in protein expression on flux through a complex synthetic pathway. Several factors influence the accumulation of proteins, including choice of promoter, ribosome binding site (RBS), gene copy number and operon structure. Furthermore, the stability of transcripts and the rate of protein degradation will naturally vary and can also be engineered (*Zelcbuch et al., 2013*).

The field of cyanobacteria-based synthetic biology has recently been attracting significant attention. However, compared to *E. coli*, studies on synthetic pathway optimization are scarce in cyanobacteria. Several review articles have presented recent advancements and discussed obstacles in the development of tools for better control of protein expression (*Angermayr, Rovira & Hellingwerf, 2015*; *Carroll et al., 2018*; *Gao et al., 2016*; *Lai & Lan, 2015*; *Ramey et al., 2015*; *Wang et al., 2012*; *Yu et al., 2013*). Most attention on the genetic parts used to engineer cyanobacteria has so far been directed toward the optimal choice of promoter (*Guerrero et al., 2012*; *Huang et al., 2010*; *Markley et al., 2015*; *Qi et al., 2013*; *Ruffing, Jensen & Strickland, 2016*) or RBS sequence (*Englund, Liang & Lindberg, 2016*; *Heidorn et al., 2011*; *Oliver et al., 2014*; *Veetil, Angermayr & Hellingwerf, 2017*; *Wang et al., 2018*; *Yunus & Jones, 2018*). The impact of gene order in a cyanobacterial synthetic operon has also been investigated (*Nozzi & Atsumi, 2015*). Other approaches have included the development of broad-host range expression systems for cyanobacteria (*Taton et al., 2014*) and evaluation of integration at different genomic loci *vs.* episomal expression (*Angermayr et al., 2014*; *Guerrero et al., 2012*). Despite these efforts, we are not aware of any prior studies that have directly quantified heterologously expressed proteins in cyanobacterial hosts. Hence, there is still limited data to determine cause-and-effect relationships in previous optimization studies.

In the present work we have focused on optimizing an engineered ethanol pathway introduced into *Synechocystis* sp. PCC 6803 via a chromosomally integrated bicistronic artificial operon. We evaluated the impact of variant RBS strengths within the operon, the choice of operon design, and previously reported modifications to native metabolism on product yield. The strains were characterized by direct measurements of both product and enzyme concentrations. Although no major advancements in product yield were observed, the study contributes to our understanding of factors that influence the outcome of metabolic engineering designs in cyanobacteria.

## METHODS

### Culture conditions

The glucose-tolerant wild-type strain of *Synechocystis* sp. PCC 6803 (provided by Dr. Patricia Armshaw, UL, Limerick, Ireland) and transformed strains thereof were were

cultured in BG-11 medium (*Stanier et al., 1971*). Cultures were grown in 100 mL Erlenmeyer flasks containing 20 mL of BG-11 at 30 °C in a Photon Systems Instruments AlgaeTron230 growth chamber having 1% (v/v) $CO_2$ (using Ecotechnics (UK) $CO_2$ sensor and controller) and a light intensity setting of 60 µmol photons/m$^2$/s (cool-white LED with added far-red LEDs 735 nM). The orbital Unimax 1010 (Heidolph Instruments) shaker was set to shake at a speed of 210 rpm. Cultures were supplemented with spectinomycin (50 µg/mL final concentration) and IPTG (one mM final concentration) when appropriate. Growth was monitored by following optical density (OD) via absorbance at 730 nm (Infinite 200 Pro spectrophotometer; Tecan Group Ltd., Zurich, Switzerland). *E.coli* strains were grown at 37 °C in Luria-Broth medium supplemented with antibiotic when necessary at the following final concentrations: spectinomycin 50 µg/mL, kanamycin 50 µg/mL or ampicillin 100 µg/mL.

### DNA constructs

All DNA manipulations were performed using BASIC DNA assembly (*Storch et al., 2015*) and plasmids transformed and propagated in *E. coli* DH5 alpha. Briefly, DNA parts used for the assembly were amplified using primers containing BASIC Prefix and Suffix and cloned into pJET1.2 vector. The primers used are listed in Table S1. The DNA constructs were assembled using BsaI-digested pJET-storage vectors and appropriate linkers as described in (*Storch et al., 2015*). BASIC linkers that encoded five different RBS sequences (Table S2) were used to assemble both single gene and operon constructs of varying translational strength. Table S3 contains a list of plasmids constructed in this study.

### *Synechocystis* sp. PCC 6803 transformation

Transformation of S*ynechocystis* sp. PCC 6803 with plasmid DNA constructs was performed as described in (*Armshaw et al., 2015*) except for the use of glucose. Spectinomycin at a final concentration of 50 µg/mL was used as a selection marker. Chromosomal integration and full segregation were verified by PCR and Sanger sequencing of the obtained amplification products (Source BioScience, UK).

### Fluorescence measurements

For eYFP measurements cells were grown up as described previously to $OD_{730\ nm}$ of 0.5 and IPTG was added to a final concentration of one mM. After 7 days, the cultures were diluted 10 times and absorbance and fluorescence readings were taken using the Infinite 200 Pro spectrophotometer (absorption: 730 nm; excitation 503 nm, emission 540 nm). The fluorescence was normalized against wild-type strain control and cell density.

### Ethanol measurements

Cells were grown up to $OD_{730\ nm}$ of 0.5 and IPTG was added to final concentration of one mM. After 7 days, one mL of every culture was taken, the cells were spun down (4,000 g, 10 min) and the ethanol content of the supernatant quantified by HPLC (1,200 series; Agilent Technologies, Böblingen, Germany). A prepacked HPLC carbohydrate analysis AMINEX HPX-87H column (300 × 7.8 mm, hydrogen form, nine µm particle size, 8% cross-linkage) with industrial grade guard cartridge (30 × 4.6 mm, hydrogen form) (Bio-Rad, Hercules,
CA, USA) was heated up to 60 °C. The mobile phase was 5 mM sulphuric acid in water (HPLC grade) with a flow rate of 0.6 mL/min. The signals were acquired with a refractive index detector. A standard curve made up of serial dilutions of absolute ethanol (VWR) was used to determine the quantity of ethanol in each sample.

## Sample preparation for SRM Mass spectrometry analysis

Cells were grown up to $OD_{730\ nm}$ of 0.5 and IPTG was added to final concentration of one mM. After 7 days, 15 mL of cultures were harvested by centrifugation (4,000 g, 30 min). The pellets were re-suspended in one mL of extraction buffer (20 mM Tris-HCl pH 8.0, one mM EDTA, two mM DTT). Five hundred microliters of cell suspension were mixed with 500 μL of extraction buffer and washed glass beads (Sigma) in a two mL lysis tube. The cells were disrupted for 10 min at 30 Hz using the TissueLyser II (Qiagen, Valencia, CA, USA). To remove unbroken cells from the lysate the samples were spun down at 12,000 g for 10 min. Protein concentration in the supernatant was estimated using the DC Protein Assay (Biorad, USA). Next, 100 μg of protein was reduced with 10 mM DTT in 50 mM ammonium bicarbonate (final concentrations) for 1 hr at 56 °C, shaking followed by alkylation with 50 mM iodoacetamide (final concentration) for 30 mins at 37 °C in the dark, 500 rpm (Eppendorf thermomixer). Proteomics-grade trypsin (Promega, USA) was added to the protein sample in a 1:50 mass ratio and the samples were incubated for about 4 h at 37 °C (700 rpm, Eppendorf thermomixer). Subsequently, trypsin was added to the samples again in the same ratio and the samples were incubated at 37 °C (700 rpm, Eppendorf thermomixer, Germany) for a further 16 hrs to ensure full digestion of the proteins. The digestion was then stopped by lowering the pH to less than 2 by adding formic acid and incubating for 30 min at 37 °C, 500 rpm. Sample clarification and removal of water-immiscible products were achieved by centrifugation for 10 min at 12,000 g. Supernatants were stored at −80 °C until further SRM analysis.

## Analysis of protein by LC-MS/MS

The LC-MS/MS comprised of an Agilent 1100 LC system and an ABSciex 6500 Qtrap MS. Chromatography was performed on a Phenomenex Luna C18(2) column (100 mm × two mm × three μm) at a temperature of 50 °C utilising a gradient solvent system of A (94.9% $H_2O$, 5% $CH_3CN$, 0.1% formic acid,) and B (94.9% $CH_3CN$, 5% $H_2O$, 0.1% formic acid). A gradient from 0% B to 35% B over 30 min at a flow rate of 250 mL/min was used. The column was then washed with 100% B for 3 min and then re-equilibrated with 100% A for 6 min. Typically 40 μL injections were used for the analysis.

The MS was configured with an Ion Drive Turbo V source, Gas 1 and 2 were set to 40 and 60 respectively, the source temperature to 500 °C and the ion spray voltage to 5500 V.

The MS, configured with high mass enabled, was used in "Trap" mode to acquire Enhanced Product Ion (EPI) scans for peptide sequencing and "Triple Quadruple" mode for Multiple Reaction Monitoring. Data were acquired and analysed using SCIEX software Analyst 1.6.1 and MultiQuant 3.0.

### Proteotypic peptides

Signature peptides for pyruvate decarboxylase (Pdc), acetaldehyde reductase (Adh, also sometimes referred to as 'alcohol dehydrogenase'), pyruvate kinase (Pyk), bifunctional fructose-1,6-bisphosphatase/sedoheptulose-1,7-bisphosphatase (Bibp) and AtpB protein (Slr1329) were determined from trial MRM-MS runs. AtpB was used as an internal standard for protein normalisation as it exhibited minimal fluctuation across treatments in a previous study (*Vuorijoki et al., 2016*). The typical work flow to select the best signature peptides consisted of samples analysis using transitions generated by an *in silico* analysis using Skyline (*MacLean et al., 2010*). The identity of candidate peptides was then confirmed by Enhanced Product Ion scans. *Synechocystis* sp. PCC 6803 background proteome (http://genome.kazusa.or.jp/cyanobase) was used to confirm the uniqueness.

Typically 3–5 transitions per peptide were used. In the final method 2–4 peptides per proteins were used for its identification and quantification. Signature peptides for each protein are listed in Table S4. Protein quantification was performed based on relative peak intensities of the analysed protein and normalized to the relative peak intensities of the AtpB native standard.

## Statistical analysis

For all statistical analysis with samples taken from microbial cultures, a normal distribution was assumed, as discussed in *Fay & Gerow (2013)*. Three biological replicates were employed for every treatment and/or condition. One-way ANOVA was carried out using the one-way ANOVA with post-hoc Tukey HSD (Honestly Significant Difference) test calculator prepared by Navendu Vasavada (http://astatsa.com/OneWay_Anova_with_TukeyHSD/). For the analysis where two factors (RBS in front of PDC, RBS in front of ADH) were evaluated in a combinatorial design we employed the anova2 and multcompare functions of Matlab R2017b. An Excel template provided by *Weissgerber et al. (2015)* was modified and used for all scatterplots.

## RESULTS AND DISCUSSION

Operon architectures, ubiquitous features of bacterial genomes, are an effective and highly efficient means for driving coordinated production of proteins involved in metabolic pathways. Operons are therefore also highly attractive for the synthetic implementation of entire heterologous metabolic pathways to enable bioproduction of target compounds (*Akhtar & Jones, 2008*). We were interested in understanding further what the best design for such operons may be when used in cyanobacteria.

For this purpose we used a well-studied and simple system, pyruvate-derived ethanol biosynthesis (*Deng & Coleman, 1999*), as a model two-protein metabolic production system. Principles derived from this simple model may be applicable to other, more complex bioproduction systems. Many other factors than pathway composition will also influence the metabolic outcome under laboratory conditions, including light and gas exchange, and when the conditions are sub-optimal this may limit the impact of improvement in any pathway. For the sake of simplicity, the focus in the present study

was placed on a relative comparison employing only modulation in pathway proteins as experimental treatments.

## Selected RBS variants are functional *in vivo* and display a wide range of protein expression strength

A total of 5 RBS variants (Table S2) were selected for characterisation via assays of fluorescence emitted by RBS-*eyfp* constructs. Four RBS sequences have been characterised previously in *E. coli*, while the fifth is specific to the anti-Shine-Dalgarno sequence of the *Synechocystis* sp. PCC 6803 ribosome (*Heidorn et al., 2011*). These sequence variants were contained within linker sequences ("RBS linkers") used in the BASIC DNA construct assembly process (*Storch et al., 2015*).

Five constructs were generated for PA1lacO1 promoter-driven expression of eYFP containing one of five 5′-proximal RBS variants (Table S3). The PA1lacO1 promoter was previously shown to be inducible with IPTG in *Synechocystis* sp. PCC 6803 (*Guerrero et al., 2012*). All ethanol pathway encoding constructs were designed for chromosomal integration at the well-studied *slr0168* neutral site (*Angermayr, Paszota & Hellingwerf, 2012*).

All five constructs were introduced into *Synechocystis* sp. PCC 6803 and fully segregated mutants were isolated. The fluorescence of the cultures was measured following a 7-day induction period in the presence of IPTG (Fig. 1). RBS A, which has been reported as weak in *E. coli* (*Salis, Mirsky & Voigt, 2009*), turned out to also show minimal expression in the cyanobacterial host strain. The commonly used RBS elements in iGEM projects, RBS B (BBa_B0064) and RBS C (BBa_B0034), displayed intermediate strength. RBS E, which has been chosen as a cyanobacteria-specific RBS, enabled the highest expression of all tested sequences. Based on the eYFP measurements, the tested RBS sequences could be placed in the following relative strength order in cyanobacteria: RBSA <RBS B <RBS D <RBS C <RBS E. This set of RBS sequences satisfied the main objectives with this initial screening, to be capable of modulating protein abundance across a broad range in cyanobacteria.

## Creation of a combinatorial library of ethanologenic strains

We wanted to evaluate how the expression and organisation of the *pdc* and *adh* genes in an operon would impact both flux and product formation through the synthetic ethanol pathway. Further, we wanted to correlate the output to protein levels and understand the impact on growth. To achieve this we designed an operon matrix with varying levels of expression of the two genetic components governed by the characterised RBS linker sequences used in the construction. In total, 25 constructs were created with all combinations of RBS sequences upstream of the *pdc* and *adh* genes. Promoter, terminator and genomic integration locus were the same as for the eYFP constructs described above. Twenty-four out of the 25 constructs were successfully used to transform *Synechocystis* followed by segregation through serial passage and confirmation by PCR. These mutants were then used to test ethanol production efficiency and quantify the abundance of the pathway components for each combination of RBS variants. The mutant containing RBS D upstream of *pdc* and RBS A upstream of *adh* could not be segregated despite repeated attempts.

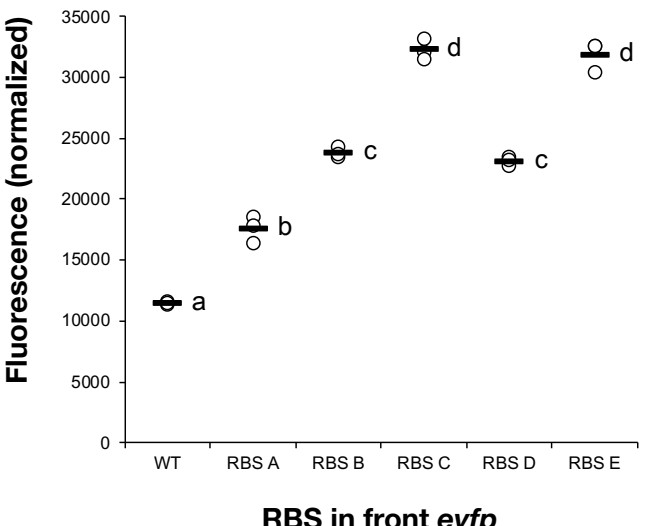

**Figure 1  Measured fluorescence indicating eYFP expression levels in *Synechocystis* sp. PCC 6803.**
Comparison of eYFP fluorescence in *Synechocystis* sp. PCC 6803 depending on the RBS sequence preceding the eYFP-encoding gene. Five different RBS sequences were investigated (A, B, C, D, E). The fluorescence was normalized against wild-type strain control and cell density. The horizontal line represents the mean average for each treatment, $n = 3$ (biological replicates). Means without a common superscript letter differ ($P < 0.05$) as analyzed by one-way ANOVA and the TUKEY test (*Assaad et al., 2014*).

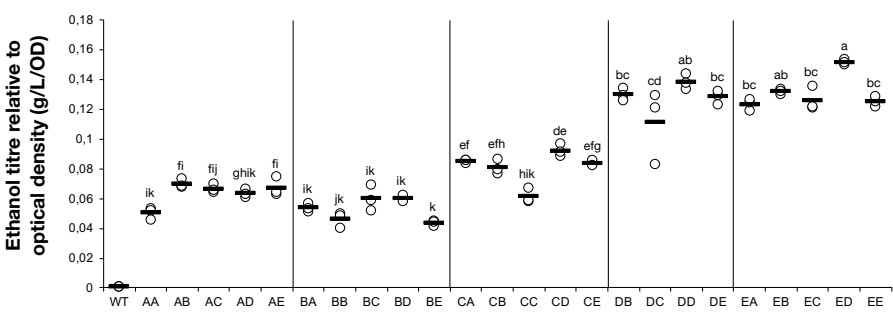

**Figure 2  The ethanol titre in cultures of *Synechocystis* sp. PCC 6803 RBSx strains.** The titre of ethanol relative to the optical density (g/L/OD) was measured after 7 days of cultivation of the 24 strains of *Synechocystis* sp. PCC 6803 with combinations of *pdc* and *adh* genes preceded by five different ribosome binding sequences (A, B, C, D, E). The plot shows individual measurements for all 24 strains and the wild-type strain. The horizontal line represents the mean average for each treatment. All ethanol titre values were normalized relative to each respective final OD. Means without a common superscript (A) or lowercase (B) letter differ ($P < 0.05$) as analyzed by two-way (1st RBS, 2nd RBS) ANOVA and the TUKEY test (*Assaad et al., 2015*).

## RBS variation results in substantial differences in ethanol production

The accumulation of ethanol after 7 days of cultivation was assessed by HPLC. As expected, variation in ethanol production was observed among the tested strains (Fig. 2). The strains exhibited minor differences in growth curves and final optical density values with the

exception of strains harboring RBS B for PDC and RBS C for ADH (Data S2). Ethanol production was strongly dependent on the RBS sequences for the *pdc* gene as significant variation in ethanol production was observed between strains harboring different RBS upstream of *pdc* while having the same RBS upstream of *adh*. No such trend was observed for variation of the RBS upstream of *adh*, when the *pdc* RBS was held constant. Interestingly, RBS A placed upstream of *pdc* led to higher ethanol production than RBS B upstream of *pdc*, even though RBS A was the weakest RBS in the context of *eyfp*. RBS D, which was considered weaker than RBS C when tested with the *eyfp* gene, provided a higher level of ethanol when placed upstream of *pdc* than RBS C. However, such differences in RBS activity are not surprising as it is well known that the strength of a chosen RBS sequence depends not only on its own sequence but also on the local DNA sequence context (*Salis, Mirsky & Voigt, 2009*). Therefore, we cannot assume that changes in RBS will lead to the same variation in protein abundance when coupled with *pdc* or *adh* as compared to *eyfp*.

## RBS variants modulate protein abundance

To understand the cause-and-effect relationships better, we analysed the abundance of Pdc and Adh in our ethanologenic strains via selected reaction monitoring (SRM) on an LC-MS/MS (Fig. 3), assuming that each internal enzyme activity at least to some degree correlated with the measured quantity of each protein. Protein abundances were normalised to the abundance of the ATP synthase beta subunit (AtpB, *slr1329*). From these data, it is clear that RBS strengths can be ranked in the following order: RBS A $\leq$ RBS B <RBS C <RBS D <RBS E. This relationship holds for both Pdc (Fig. 3A) and Adh (Fig. 3B). When a weak RBS (RBS A or RBS B) was placed upstream of *pdc*, regardless of the RBS present upstream of *adh*, both the ethanol yield and relative abundance of Pdc were low. With increasing *pdc* RBS strength the production of ethanol and the accumulation of both Pdc and Adh all increased. The strength of the RBS upstream of *adh*, while capable of modulating Adh accumulation, had negligible influence on the abundance of Pdc.

Next, we calculated the correlation between enzyme abundance and ethanol yield and also compared how the abundance of one enzyme correlated with the other. The correlation between protein abundance and ethanol yield was strongest for Pdc ($R^2 = 0.89$; Fig. 4A), while for Adh this correlation was much weaker ($R^2 = 0.44$; Fig. 4B). Comparing protein levels, Adh abundance was, although weak, slightly more strongly correlated with Pdc abundance ($R^2 = 0.51$) (Data S2, Fig. 4 sheet). Together, these observations indicate the operation of "translational coupling" (*Levin-Karp et al., 2013*) between high-level Pdc expression and Adh.

## Enhanced Pdc expression is sufficient for yield-improvement

The weak correlation between Adh and ethanol titre suggested that Adh over-expression was not necessary. In order to test this hypothesis, five new strains were constructed that were similar to the previous 24 strains except that they lacked the *adh* gene. Once again, five different RBS sequences were placed upstream of *pdc*. The constructs were transformed into *Synechocystis* 6803 and fully segregated mutants were obtained for all combinations. Ethanol production and protein abundance were tested, with five of the

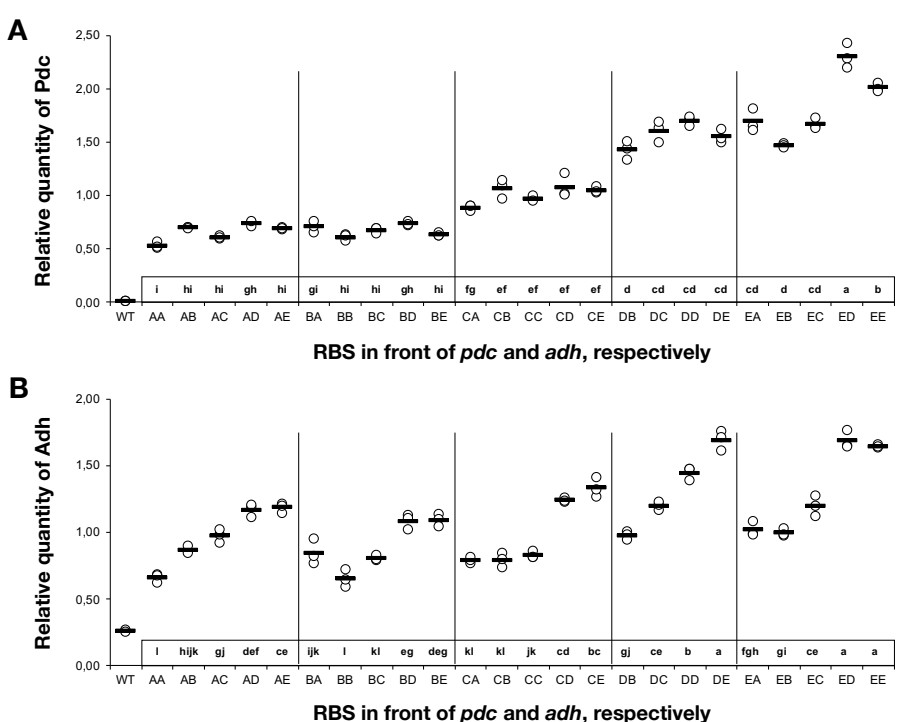

**Figure 3** **The relative quantity of Pdc & Adh protein in *Synechocystis* sp. PCC 6803 RBSx ethanol producing strains after 7 days of cultivation.** (A) Pdc and (B) Adh. The $Y$-axis displays the relative quantity of Pdc or Adh when normalised to the internal standard, AtpB. The horizontal line represents the mean average for each treatment. On the $X$-axis, the first capitalized letter stands for the RBS upstream of the *pdc* gene, the second capitalized letter stands for the RBS upstream of *adh*. The inset just above the $X$-axis displays grouping of means by a common lowercase letter that do not differ ($P < 0.05$) as analyzed by two-way ANOVA and the TUKEY test (*Assaad et al., 2015*). WT = wild-type *Synechocystis* sp. PCC 6803.

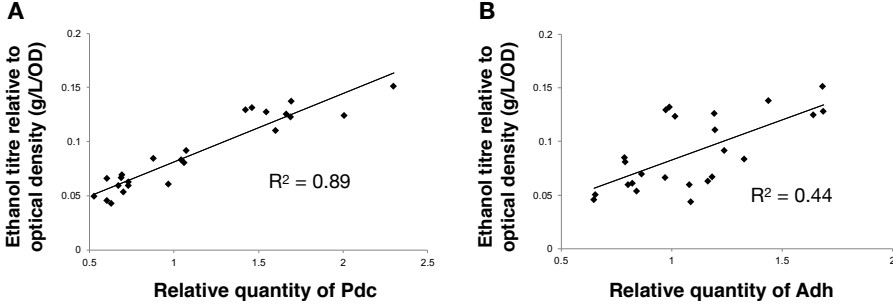

**Figure 4** **The relationship between the ethanol titre and the relative quantity of each pathway protein in *Synechocystis* sp. PCC 6803 RBSx ethanol producing strains after 7 days of cultivation.** (A) Pdc and (B) Adh. The $R^2$ value was obtained from a linear fitted regression using Excel. The same data is plotted also in Figs. 2 and 3.

previously constructed ethanol-producing strains used as controls (AA, BB, CC, DD, EE). Ethanol yields were, greater for the monocistronic (Pdc-only) strains compared with the equivalent RBS bicistronic (Pdc-Adh) strains, when comparing C, D and E with CC, DD and EE, respectively (Fig. 5A). As expected, the Pdc protein abundance was also greater in three out of five mono- vs. double-cistronic comparisons (i.e., A, D, E all had increased Pdc content compared to AA, DD and EE, respectively; Fig. 5B). These observations are consistent with either expression from a shorter transcript being more efficient and/or that there is a reduced metabolic burden placed on the cell as fewer resources are required to express the single protein. Alternatively, the new transcripts may have differed in their secondary structure. *Lim, Lee & Hussein (2011)* have reported that, in *E. coli*, proteins encoded on long operons are more highly expressed. This is in contrast to what we observed in cyanobacteria with proteins encoded on longer operons being poorly expressed. Pdc abundance was once again linearly correlated with ethanol yield (Fig. 5C) and can be improved further as there is no evidence of a plateau in ethanol productivity at the protein abundances achieved in these strains.

## Further modulation of host metabolism does not improve product yield

Sustainable commercial production of ethanol using cyanobacteria will require increased productivities and product titres. The strong relationship between ethanol productivity and quantity of pyruvate decarboxylase suggested this was the key limiting factor. However, previous studies have found at least some impact from optimization of native metabolism even when the activity of a single pyruvate-dependent enzyme (lactate dehydrogenase) was clearly rate-limiting (*Angermayr et al., 2014*). We therefore evaluated the effect of overexpressing pyruvate kinase (Pyk) from *E. coli*, which should improve the supply of the precursor pyruvate as was previously reported (*Angermayr et al., 2014*). Another possibility is to over-express BiBP, a bifunctional fructose-1,6-bisphosphatase/sedoheptulose-1,7-bisphosphatase from *Synechocystis* sp. PCC 6803 (*Jiang, Wang & Wen, 2012*), which has been reported to improve growth and photosynthetic rate in the cyanobacterium *Synechococcus* sp. PCC 7002 and *Synechocystis* sp. PCC 6803 presumably by enhancing flux through central carbon metabolism (*De Porcellinis et al., 2018*; *Liang et al., 2018*).

To strains already harbouring the *pdc-adh* cassette with RBS E in front of both genes, we introduced Pyk and BiBP as a second operon at a different genomic locus (*slr1395*). The new cassette harbouring Pyk and BiBP was combined with four different RBS elements (A, B, D, E) located upstream of each coding sequence and placed under the control of the constitutive $P_{trc}$ promoter. Fully segregated mutants were obtained for eight out of sixteen designs. However, analysis of these strains revealed no improvement in ethanol production (Fig. 6A) compared to the original strain, despite detectable increases in both Pyk and BiBP abundance (Fig. 6B). In an earlier study with *Synechococcus* sp. PCC 7002, over-expression of BiBP resulted in a growth benefit dependent on both high light and high carbon availability (*De Porcellinis et al., 2018*). In the present study, the over-expression of BiBP had no or a negative impact on growth (Fig. 6C). However, the environmental conditions, the engineering designs and the host strain were different in the two studies,

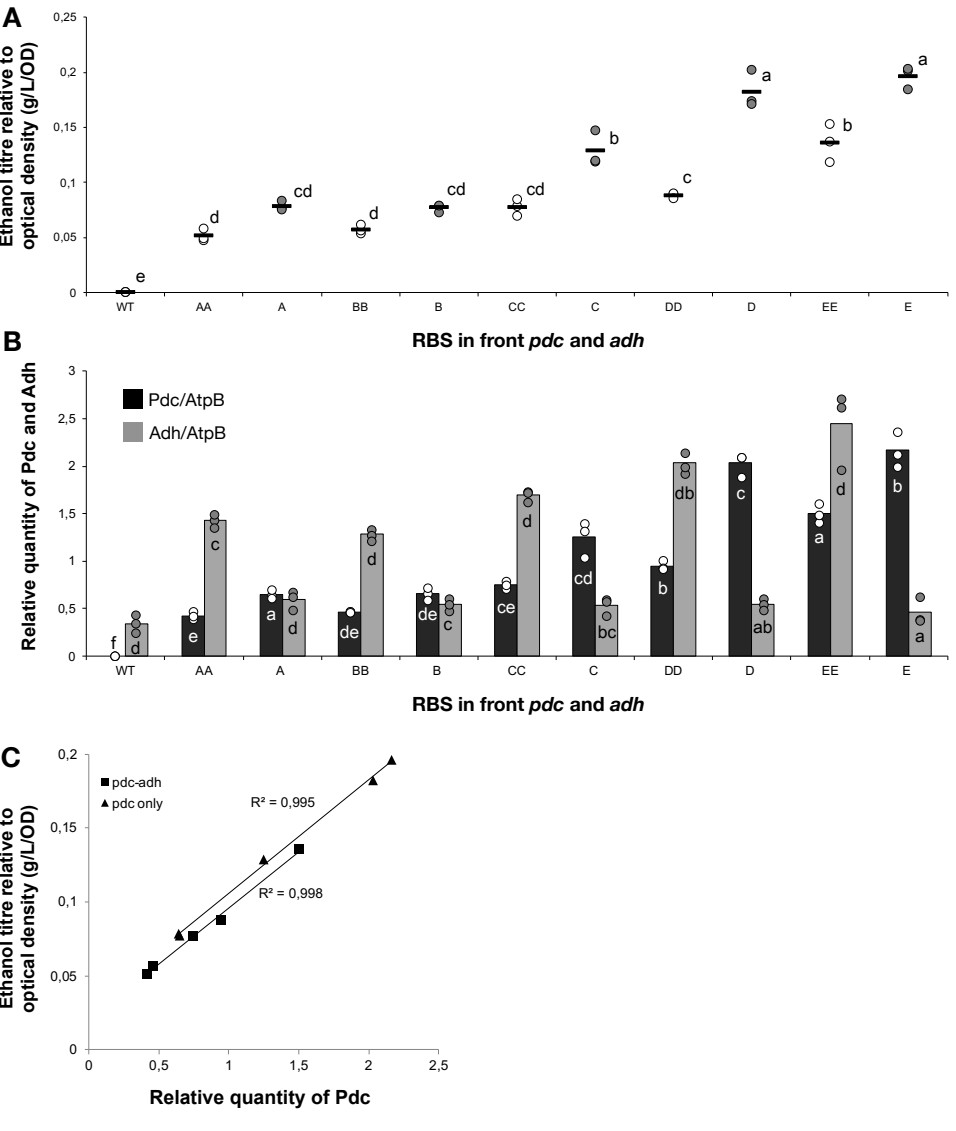

**Figure 5** **The ethanol titre and relative quantity of ethanol pathway proteins in mono- and bi-cistronic ethanologenic strains.** The experimental treatments were 5 strains with the same ribosome binding sequence in front of both *pdc* and *adh* genes (AA, BB, CC, DD, EE) and 5 strains of *Synechocystis sp.* PCC 6803 harbouring only the *pdc* gene preceded by one of five different ribosome binding sequences (A, B, C, D, E). The strains were cultured for 7 days before samples were taken for measurements. (A) Ethanol titre. (B) Comparison of the expression levels of Pdc (black bars) and Adh (grey bars), normalised relative to AtpB, in *Synechocystis* sp. PCC 6803 strains harboring the whole ethanol cassette or the *pdc* gene alone. The horizontal line (in A) or bar (in B) represent the mean average for each treatment. Means without a common lowercase letter differ ($P < 0.05$) as analyzed by one-way ANOVA and the TUKEY test (*Assaad et al., 2014*). One-way ANOVA was employed instead of two-way ANOVA in case the expression of Pdc was affected by the presence or absence of *adh*. (C) Correlation between ethanol titer and quantity of Pdc. On the *X*-axis of panels (B) and (C), the first capitalized letter stands for the RBS upstream of the *pdc* gene, the second capitalized letter (if present) stands for the RBS upstream of *adh*. All ethanol titre values were normalized relative to each respective final OD.

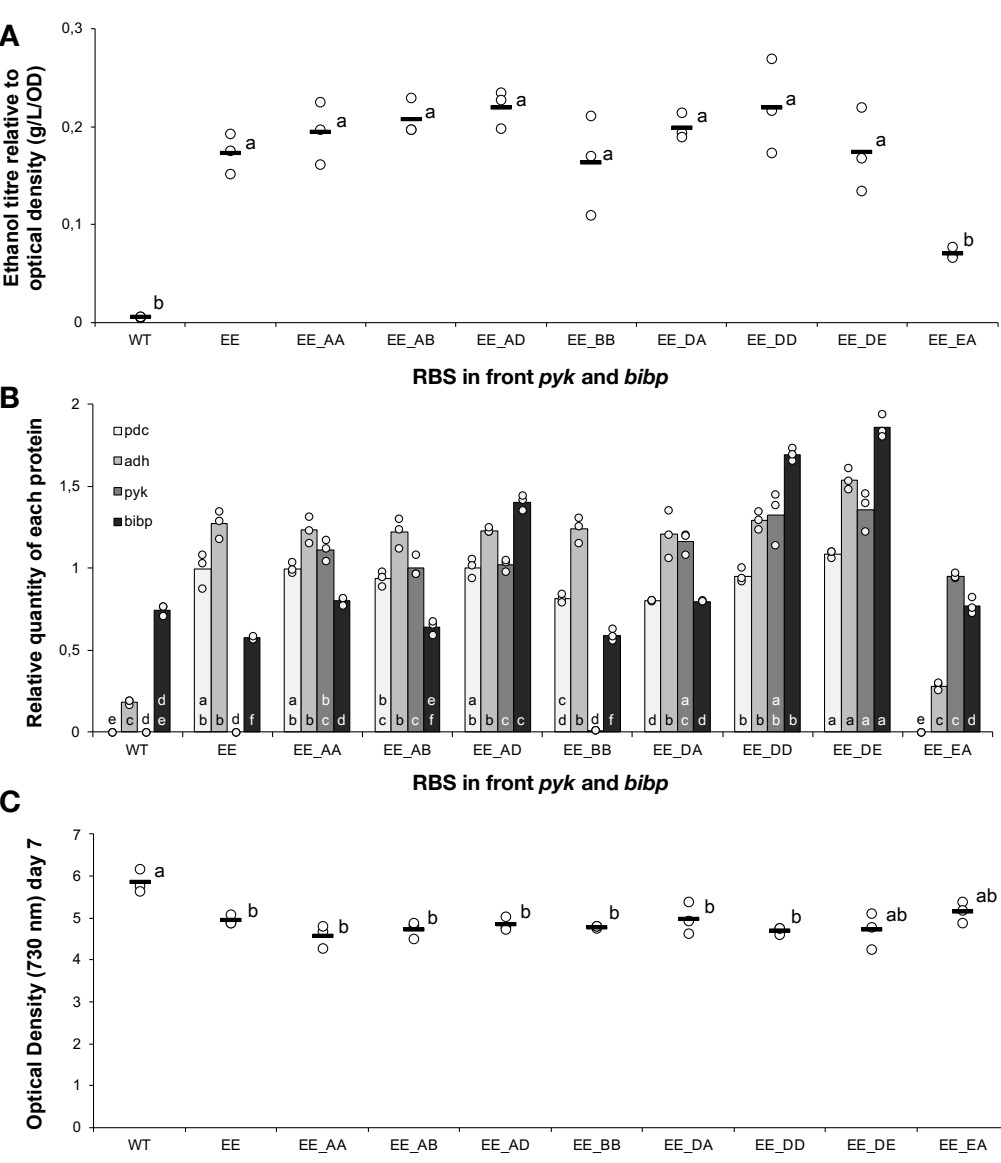

**Figure 6** **The ethanol titre, relative quantity of introduced proteins and final optical density in strains with modified central carbon metabolism enzymes.** Apart from the wild-type control (WT), all strains were based on *Synechocystis sp.* PCC 6803 with an operon containing *pdc* and *adh*, both preceded by ribosome binding site E, integrated into the slr0168 neutral site. The experimental treatments consisted of the presence or absence of a second operon located at a different genomic locus (slr1395), in which five different ribosome binding sequences (A, B, C, D, E) preceded both *pyk* and *bibp*- note that strains could only be obtained with some of the combinations. The strains were cultured for 7 days before samples were taken for measurements. (A) Ethanol titre. (B) Comparison of the expression levels of Pdc (light grey), Adh (medium grey), Pyk (dark grey) and BiBP (black) normalised relative to AtpB. (C) Final optical density on day 7. The horizontal line (in A and C) or bar (in B) represent the mean average for each treatment. Means without a common lowercase letter just above the *X*-axis differ (P¡0.05) as analyzed by one-way ANOVA and the TUKEY test (*Assaad et al., 2014*). Independent ANOVA tests were carrie out for each of the four proteins. All ethanol titre values were normalized relative to each respective final OD.

thereby ruling out any possibility to compare between them. Nevertheless, the increase in Pyk and BiBP abundance correlated well with the strength of the RBS used. Again we observed that the increased expression level of the protein encoded first in the operon (*pyk*) influenced the expression of the protein encoded last (*Bibp*). For example, this can be seen if we compare the expression of Pyk and BiBP in strains EE_AD & EE_DD.

## New insights were uncovered and several challenges remain to be addressed

All DNA constructs used in this study were generated via BASIC methodology (*Storch et al., 2015*). This method enabled quick and reliable generation of a series of DNA constructs. The main advantage of BASIC is its modularity, where a DNA part once cloned into a storage vector can be used in various assemblies without any further need for amplification. It proved extremely useful in our work involving the preparation of a large number of molecular constructs with different genetic parts, including small RBS elements. In overlap assembly techniques like Gibson (*Gibson et al., 2009*) such work would involve amplification of each gene with different primers for each construct in order to introduce the RBS sequence and compatible overhangs. Whereas when using BASIC the order of assembly is directed by the employed linkers which may also contain the RBS sequence, for example. Furthermore, in contrast to Golden Gate based approaches, the single-tier BASIC formatted parts can be assembled in any order and in varying numbers without reformatting. The approach we took significantly reduced the workload and enabled us to generate a large number of the required molecular constructs in a very short time. The bottleneck in strain construction was instead shifted to cyanobacteria transformation and segregation. For example, we were unable to isolate a strain containing RBS D upstream of *pdc* and RBS A upstream of *adh*. Frequently mooted hypotheses for such observations include metabolic burden of protein over-production and product or intermediate substrate toxicity. We infer from our data that this strain would not have produced higher levels of protein, ethanol or the intermediate acetaldehyde than other strains in the set. It would appear therefore, that some other underlying phenomenon may be at play.

We could also not isolate any strains for the four-cistron operon designs and several attempts were required to generate strains containing tri-cistronic operons. This suggests that there exists a trade-off between integration/segregation efficiency and insert length. To the best of our knowledge there have been no detailed studies performed on the impact of insert length on transformation efficiency in cyanobacteria. Three tri-cistronic operon strains remained only partially segregated, which cannot be confidently attributed to genomic integration locus, burden of protein over-production or substract/product toxicity. Operon lengths in these cases were identical. The presence of specific sequence features within these constructs and their impact on genome stability cannot be discounted (*Jones, 2014*).

In terms of lessons learned, one of the more striking observations in our data is the evidence for translational coupling. Translational coupling in general is not well characterised, particularly in cyanobacteria. The extent of translational coupling is related

to inter-cistronic distance and has been at least partially characterised in other species (*Levin-Karp et al., 2013*). Despite this being highly relevant for strain design, the distances required to either maximise or abolish translational coupling in cyanobacterial species remains entirely undescribed. This also highlights that, while assembly methods such as BASIC are extremely useful tools for rapid prototyping, other bottlenecks may limit throughput and delivery of fully optimized bioproduction strains.

It has been reported by others that longer operons result in higher protein abundance for proximally-encoded cistrons (*Lim, Lee & Hussein, 2011*). Their results, demonstrated using *E. coli*, are not consistent with the data we present here on Pdc abundance: longer operons result in lower protein abundance for proximally-encoded cistrons in *Synechocystis* sp. PCC 6803. Whether our observations are specific for the ethanol-cassette design or reflect more general differences between the cyanobacteria and other model bacterial species remains to be investigated.

The work also illustrates the importance of limiting factors for metabolic engineering, as also highlighted in the work of Angermayr and colleagues with lactate (*Angermayr & Hellingwerf, 2013*). At the enzyme level, the first enzyme of the introduced pathway (Pdc) was clearly limiting flux through the pathway and changes to the second enzyme (Adh) did not have much of an impact. At the whole-cell metabolism level, despite successful over-expression of two enzymes previously demonstrated to stimulate either growth or pyruvate-dependent product yield, no or minimal impact on the ethanol pathway was observed. If possible, identification of the key limiting factor(s) that control most of the metabolic flux through an introduced pathway, should first be pursued. With respect to the ethanol pathway, Pdc was that factor. Future studies should investigate how this activity could be enhanced.

## CONCLUSION

Using ethanol production as a model system, we have learned more about some of the considerations required for implementing optimal synthetic metabolic strain designs in *Synechocystis* sp. PCC 6803. (1) RBS variation in operon constructs can result in at least a 3.5-fold variation in pathway activity. (2) Translational coupling can have a major influence on the expression of multiple proteins encoded by an operon, e.g., influencing choice of gene order. (3) The presence or absence of downstream genes in an operon (i.e., operon length and/or composition) can also influence the expression of upstream genes. And (4), BASIC is a useful methodology for rapid assembly of construct libraries with diversified composition.

### Funding

This project has received funding from the European Union's Seventh Framework Programme FP7 (DEMA) project no. 309086. The funders had no role in study design, data collection and analysis, decision to publish, or preparation of the manuscript.

## Grant Disclosures

The following grant information was disclosed by the authors:

European Union's Seventh Framework Programme FP7 (DEMA) project: 309086.

## Competing Interests

The authors declare there are no competing interests.

## Author Contributions

- Paulina Bartasun conceived and designed the experiments, performed the experiments, analyzed the data, prepared figures and/or tables, authored or reviewed drafts of the paper, approved the final draft.
- Nicole Prandi, Marko Storch, Yarin Aknin, Mark Bennett and Arianna Palma performed the experiments, analyzed the data, authored or reviewed drafts of the paper, approved the final draft.
- Geoff Baldwin and Yumiko Sakuragi conceived and designed the experiments, authored or reviewed drafts of the paper, approved the final draft.
- Patrik R. Jones conceived and designed the experiments, analyzed the data, prepared figures and/or tables, authored or reviewed drafts of the paper, approved the final draft.
- John Rowland analyzed the data, authored or reviewed drafts of the paper, approved the final draft.

## Data Availability

The raw data is available in the Supplemental Files. The data for each figure is provided as a separate tab.

## Supplemental Information

Supplemental information for this article can be found online at http://dx.doi.org/10.7717/peerj.7529#supplemental-information.

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
