# Peer review of "The effect of modulating the quantity of enzymes in a model ethanol pathway on metabolic flux in Synechocystis sp. PCC 6803"

_PeerJ, doi:10.7717/peerj.7529_

## Round 0.1 · original submission · Major Revisions

We have receive three extensive reviews which, while generally enthusiastic, suggest several areas for improvement. We look forward to receiving a revised version of your manuscript.

Reviewer 1 ·

Basic reporting

This study reports a set of experiments aimed at the assessment the effect of altering the expression level of individual enzymes of a heterologous pathway, in the metabolism of a cyanobacterium, on the level of production of the product of that pathway. The manuscript is technically well-written, and the majority of the experiments has been carried out with state-of-the-art techniques. The design of the study, however, suffers from two basic weaknesses:
1] In a study in which the effect of variation of the expression level of heterologous enzymes is investigated, the factors limiting product formation should be clearly defined. Because of the choices made, this aspect is not properly covered: In many conditions presumably the availability of light will have been the limiting factor, which then makes it obvious that a poor correlation between product level and enzyme expression level is observed.
2] As a metabolic engineering study, the most relevant property to assay – next to final product level – is the amount of active enzyme present in the cells. This amount may differ from the total amount of enzyme-protein present, due to: misfolding of the overproduced enzyme, post-translational modifications, etc. Information on this aspect is fully lacking in this study.

Experimental design

There are several technical imperfections identifiable in the manuscript. Examples are:
a) The description of the physiological experiments (Cells were grown up to OD730nm of 0.5 and IPTG was added to final concentration of 1 mM. After 7 days, 1 mL of every culture was taken, the cells were spun down (4000 g, 10 min) and the ethanol content of the supernatant quantified by HPLC……..) is insufficiently detailed and does e.g. not allow one to calculate final product titers to be calculated. Disappearance of ethanol and/or acetaldehyde from the system is not taken into account.
b) Genetic instability of the transgenic strains is not addressed, and the check of strain construction with PCR (lines 107-108) does not provide sufficient detail).
c) Several figures and aspects of figures are redundant (e.g. Fig. 2 and background colors in Fig. 3A).
d) The landmark study in this field (Gao Z, Zhao H, Li Z, Tan X, Lu X (2012) Photosynthetic production of ethanol from carbon dioxide in genetically engineered cyanobacteria. Energy & Environ Sci 5:9857. doi:10.1039/c2ee22675h) is not in the reference list. Presumably the titers in the current study are much lower than in this study.

Validity of the findings

Several single observations (on e.g. operon length, order of genes, effect of reading frame on RBS strength, etc.) are generalized without proper justification. Also, justification of the selection of AtpB as the base for normalization of enzyme expression levels is lacking.

Additional comments

My evaluation of this manuscript is that it needs a complete re-write and shortening, in which the limited number of lessons learned are clearly described.

Reviewer 2 ·

Basic reporting

The article “Lessons learned from optimization of an introduced metabolic pathway in Synechocystis sp. PCC 6803” describes an interesting principle-based design approach to assess the mutual impact of different RBS variations on gene expression from a dicistronic operon in a model cyanobacterium. The authors used a selection of 5 well-established RBS variants to drive the likewise well-established ethanol biosynthesis pathway, consisting of pyruvate decarboxylase and alcohol dehydrogenase. This work is relevant to the fields of synthetic biology and metabolic engineering (in cyanobacteria), and it introduces an important aspect of synthetic operon design for optimized and balanced gene expression in cyanobacteria.
The language of the article is largely clear and unambiguous. Some exceptions are listed below.
Regarding the literature references, a few further suggestions are given below that might also improve the discussion.
The structure of the article is professional, all figures are relevant and of good quality. Minor concerns about figure captions are listed below.
Unfortunately, I have two major concerns regarding (i) the lack of physiological/ growth data and (ii) the lack of actual (unprocessed) raw data in the manuscript/ supplementary files.
(i) In all EtOH experiments titres were normalized to the cell density (OD730), without providing the (a) voluminetric titers and (b) the optical densities (except for Fig. 7). There is also an inconsistency between the main manuscript (figures) and the supplemental data, where apparently the same values are annotated as volumetric titres (e.g. Fig.3, 5, 6, 7- Ethanol (g/l)). This would make sense, if the OD values were ~1, but they should be provided in the manuscript.
(ii) The provided data in Supplementary file 2 are not actually raw, but already processed and normalized values that correspond to the provided diagrams. For sound reproduction of the data, all actually measured values would be required.

Abstract
1. line 26: I suggest to use the more common ‘alcohol dehydrogenase’ instead of ‘acetaldehyde reductase’, and since the enzymes are named 3 times in the abstract, ADH and PDC might be used.
2. line 29 : “operon design and modifications to native metabolism on pathway flux was monitored” … sounds too ambitiously worded for what has actually been done (and monitored). Lines 40-43 should be sufficient for this aspect
3. line 38: “translational coupling can also occur in cyanobacteria.” rather: “…can be implemented by design in cyanobacteria…”

Introduction
The introduction gives a brief overview of the state of knowledge regarding the challenges and available solution in synthetic pathway design. The first passage (until line 54) might be improved by adding the aspect of intra- and intergenic ‘compositional context’ of genetic constructs, referring to the recent publications of Cambray et al., 2018 (https://doi.org/10.1038/nbt.4238) and Yeung et al., 2017 (https://doi.org/10.1016/j.cels.2017.06.001). These papers are also of relevance to this manuscript as they support the discussion of different relative protein levels using different CDS (line 245). Accordingly, in the next subsection (lines 55-63), the paper by Thiel et al 2018 (doi: 10.1186/s12934-018-0882-2.) should be cited as a work that has already addressed the influence of CDS in cyanobacteria.
4. line 50: citation format. “For example, (Zelcbuch et al. 2013) clearly…” should be “For example, Zelcbuch et al. (2013) clearly…”
Methods
While the analytical part of the methods section is very elaborate, the cyanobacterial culture conditions should be described with more details and technical specifications, since these can be very critical to reproducibility of data derived from phototrophic conditions:
5. General inconsistencies regarding µl/ml vs. µL/mL: microliter is mostly written ‘µl’, milliliter mostly ‘mL’
6. General suggestion: Full plasmid sequences (at least exemplarily) might be provided as supplement data file (e.g. genbank format) or on an open data repository
7. lines 86,90 and several time in lines 133-142: °C –> gap between number and unit sometimes missing
8. line 84: “other cyanobacterial strains ‘derived thereof’…”
9. line 84-86: more specifications of the cultivation equipment might be provided (shaker orbit, type of light source [fluorescence, diodes?])
10. line 85: BG-11 ‘media’ - > ‘medium’
11. line 86: how was 1% CO2 supplied (CO2 atmosphesre or bubble column)? Which light source (e.g. fluorescent vs. photodiodes, white light vs. specific wavelengths)? Which shaker was used?
12. line 106: final concentration ‘of’
13. line 115-123: was an internal and/or external standards used for calibration of HPLC analysis? How were the data calculated?
14. line 127: extraction buffer
15. line 127-128: 500 µL

Results and Discussion
16. lines 189-191: I suggest to briefly introduce the basic pathway “Glycolysis  Pyruvate  Acetaldehyde  Ethanol” here for synthetic biologists without background in metabolic engineering. It could also be included in Fig. 2 for a clear overview
17. also a schematic/representative plasmid map to briefly depict the vector composition and the modules could be added to Fig. 2
18. lines 198-199: “These sequence variants were encoded within linker sequences (“RBS linkers”)”
 what does ‘encoded’ mean in this context? should it rather be replaced by ‘attached to’ or ‘flanked by’?
19. line 203: “shown to exhibit some IPTG-inducibility in cyanobacteria”. Rather provide a more specific or quantitative characteristic than “some inducibility”. Also change ‘cyanobacteria’ to ‘Synechocystis sp. PCC 6803’.
20. lines 214-216: to give a quantitative idea of the ‘dynamic range’ of this RBS series, e.g. the maximal expression ratio/fold-change (RBS A vs. RBS E) should be provided
o also in the following passages (lines 233-247; lines 250-259) quantitative statements regarding the respective expression ranges would improve the informative value of the data description

21. Line 227-228: Synechocystis was “transformed” with the constructs, not vice versa
22. Lines 244-245: this discussion can be supported with more recent literature (Thiel et al., 2018)
23. Lines 263-264: “Comparing protein levels, Adh abundance was in fact more strongly correlated with Pdc abundance (R2 = 0.51).” Compared to which comparison are these protein abundances more strongly correlated?
Also: ADH-PDC protein correlation plot should be shown as well, since a clear statement is made here.
I suggest to discuss these data more carefully, since two similar R2 values are interpreted in a positive (R2 = 0.51 -> translational coupling) and a negative (R2 = 0.44 -> ADH overexpression might be redundant) way, respectively
24. Lines 278-279: what does ICL stand for?
25. Lines 284-291: also here, differential characteristics of DNA topology and/or RNA secondary structure might play a role in different expression levels
26. Line 307: slr1395 -> italics

Figures and Captions
27. In general: there are several cases, where there is a space missing between two words (e.g. Synechocystissp.) in several captions
Figure 1:
28. While the method section says that “The fluorescence was normalized
against wild-type strain control and cell density.”, there is no indication for that neither in the diagram, nor in the caption. “Fluorescence (measured)” rather indicates raw data. The y-axis label and caption should be more specific.
29. RBS in front ‘of’ eyfp

Figure 2:
30. The caption should contain more information to be self-contained

Figure 3:
31. In the methods section the construct series were named “RBSx_eYFP and RBSx_EtOH”. For consistency, I suggest to stick to this specification in this (and the other) caption(s)
32. “Mean average ethanol titres of each cultures…”  “…culture…”

Figure 5:
33. This caption should refer to Fig. 3+4

Figure 6:
34. The caption lacks the information of OD normalization, which is indicated on the Y-axis. According to the data in suppl. File 2, these values are g/L values. These inconsistencies need to be checked throughout all figures/data

Supplementary Material

Supplementary Table 1
35. Synechocystis and E. coli  italics
36. the BASIC prefix and suffix should be labeled (which is in the forward and reverse sequences?)
37. the PA1lacO1 reverse sequence is completely in bold

Supplementary Table 2
38. for ‘RBS E’ the annotation ‘RBS*’ should be added to the ‘additional information’ column (according to Heidorn et al)
39. the table caption should explain the capitalized nucleotides

Experimental design

The experimental design is overall sound and well-chosen to address the central question. The use of state-of-the-art techniques, in particular regarding cloning and protein quantification is further inspiring for follow-up projects in the field of cyanobacterial synthetic biology.
However, a major drawback of the design is a lack of physiological analysis, particularly regarding growth characteristics of the examined strains. In lines 221-222 the authors write: “Further, we wanted to correlate the output to protein levels and understand the impact on growth.” Growth characteristics are not provided, except for the final experiment, where all strains harbored the same RBS combination for PDC and ADH (Fig. 7C). And here, only endpoint data are provided.
Particularly regarding the potential acetaldehyde toxicity, there might be a bias in growth and biomass accumulation between strains with low and high ADH levels. All EtOH data are (according to the main document) normalized to optical density. If higher PDC abundance would result in higher acetaldehyde titers and potentially growth inhibition, similar (or even lower) acetaldehyde to EtOH convertion rates might result in higher apparent per-cell EtOH titers. This is certainly important for the conclusion that “abolishment of
reductase over-expression resulted in the greatest ethanol productivity”. Therefore, growth curves – or at least endpoint OD730 values – need to be disclosed in order to exclude a bias from severe growth effects.

Validity of the findings

The data are statistically well analyzed. However, there are inconsistencies regarding the normalization of EtOH titres between main document and supplement. More transparency regarding physiological data/ growth would be important to substantiate the interpretation of the results.
See also previous sections.

Additional comments

I suggest to run a replicate experiment with selected strains in order assess and discuss the physiological outcome (i.e. growth curve vs. EtOH accumulation) of the different PDC/ADH titres over time.

Reviewer 3 ·

Basic reporting

The title “Lessons learned from ..” fails to represent the manuscript and looks like that of review articles. Several keywords such as “ribosome binding site sequences”, “translational coupling”, and “operons” should be included.

Experimental design

Line 267, “The relative abundance of Pdc was generally higher than Adh regardless of which RBS was used”. The claim was not supported by the experimental data since ” Protein quantification was performed based on relative peak intensities of the analysed protein” . Please determined the expression levels of Pdc and Adh proteins by using the suitable standard peptides.

Validity of the findings

Fig. 3 The data shown in Fig 3B is not required since the results are summarized in Fig. 3A

---

## Round 0.2 · Minor Revisions

Please address the few remaining issues.

Reviewer 2 ·

Basic reporting

The manuscript has improved a lot and I am happy that the authors addressed all concerns, either by supplying the requested information/ changes, or by convincing argumentation.

However, I have a remaining concern regarding the statement in line 222-223: "The strains exhibited minimal differences in growth curves and final optical density values"
According to the added supplementary data (Fig. 2) there are actual, gradual differences of up to ~30% in the final OD730. This is not exactly 'minimal' and should be described more accurately, since the ‘impact on growth’ is clearly defined as part of the study (line 209) . In particular, strains harboring RBS B for PCD and RBS C for ADH show higher final ODs compared to the rest.

The growth curve diagram in Suppl. Fig 2 refers to data on a local drive, which is not remotely accessible (and hence no graph is visible!); I therefore suggest to provide these data in a supplementary spreadsheet or in an online repository, in accordance with the journal's 'open data' standard. Also, raw data of the ODs for the YFP normalization (Fig.1) are not yet provided.

I suggest to publish the study ASAP after some brief revision of the mentioned points.

Experimental design

OK

Validity of the findings

Underlying data not fully provided, See 'Basic reporting'.

---

## Round 0.3 · accepted · Accept

Thank your for your quick response to the final minor issues